



# Using a coupled LES-aerosol radiation model to investigate urban haze: Sensitivity to aerosol loading and meteorological conditions

Jessica Slater[1], Juha Tonttila[2], Gordon McFiggans[1], Sami Romakkaniemi[2], Thomas Kühn[2,3], and Hugh Coe[1]

[1]Centre for Atmospheric Sciences, School of Earth and Environmental Sciences, University of Manchester, Manchester, UK
[2]Finnish Meteorological Institute, Atmospheric Research Centre of Eastern Finland, Kuopio, Finland
[3]Department of Applied Physics, University of Eastern Finland, Kuopio, Finland

**Correspondence:** Hugh Coe (hugh.coe@manchester.ac.uk)

**Abstract.** The aerosol-radiation-meteorological feedback loop is the process by which aerosols interact with solar radiation to influence boundary layer meteorology. Through this feedback, aerosols cause cooling of the surface, resulting in reduced buoyant turbulence, enhanced atmospheric stratification and suppressed boundary layer growth. These changes in meteorology result in the accumulation of aerosols in a shallow boundary layer, which can enhance the extent of aerosol-radiation interac-

tions. The feedback effect is thought to be important during periods of high aerosol concentrations, for example during urban haze. However, direct quantification and isolation of the factors and processes affecting the feedback loop has thus far been limited to observations and low resolution modelling studies. The coupled LES-aerosol model, UCLALES-SALSA, allows for direct interpretation on the sensitivity of boundary layer dynamics to aerosol perturbations. In this work, UCLALES-SALSA has for the first time been explicitly set up to model the urban environment, including addition of an anthropogenic heat flux

and treatment of heat storage terms, to examine the sensitivity of meteorology to the newly coupled aerosol-radiation scheme. We find that: a) Sensitivity of boundary layer dynamics in the model to initial meteorological conditions is extremely high, b) Simulations with high aerosol loading (220 $\mu$g/m$^3$) compared to low aerosol loading (55 $\mu$g/m$^3$) cause overall surface cooling and a reduction in sensible heat flux, turbulent kinetic energy and planetary boundary layer height for all three days examined and c) Initial meteorological conditions impact the vertical distribution of aerosols throughout the day.

## 1 Introduction

Severe air pollution events are a major health issue for megacities worldwide, particularly in nations with large populations and high levels of industrialisation such as India and China. Beijing, situated in the North China Plain is well known for its air quality issues, where concentrations of PM$_{2.5}$ (particulate matter with a diameter $<$ 2.5 $\mu$m) frequently exceed the World Health Organisation's recommended hourly exposure limit of 25 $\mu$g/m$^3$. Heavy 'haze' periods envelop Beijing due to

a complex combination of emission sources and unfavourable meteorology. Observations have identified the importance of changing synoptic conditions on the onset of haze episodes, while the longevity and intensity of the episodes are found to be affected by aerosol-radiation interactions. These interactions feedback on boundary layer meteorology to cause unfavourable





conditions such as temperature inversions, increased humidity and decreased wind speed (Dou et al., 2015; Zhang et al., 2015, 2017; Wang et al., 2019; Zhong et al., 2019b).

In addition to the unfavourable meteorological conditions; heavy emissions and regional transport of pollutants into Beijing cause high concentrations of urban aerosol particles to accumulate. These particles can either scatter or absorb solar radiation, depending on their composition. Observations predominantly show that aerosol particles cause net cooling at the surface and warming in the upper atmosphere. This consequently alters the thermal profile of the atmosphere, reducing turbulence due to buoyancy. Reduced turbulent mixing suppresses boundary layer development during the day, minimises the vertical
distribution of pollutants and increases surface aerosol concentrations. Furthermore, reduction in planetary boundary layer (PBL) height due to the feedback effect also increases water vapour concentrations which can result in enhanced aqueous heterogeneous reactions, thus increasing the rate of secondary aerosol formation. If the aerosol particles are hygroscopic, increased water vapour concentrations will also cause particle growth, resulting in stronger aerosol-radiation interactions. This positive feedback loop between aerosols, radiation and meteorology can lead to sustained periods of stagnation and has been
found to enhance pollution events (Figure 1) (Liu et al., 2018b; Luan et al., 2018; Petäjä et al., 2016).

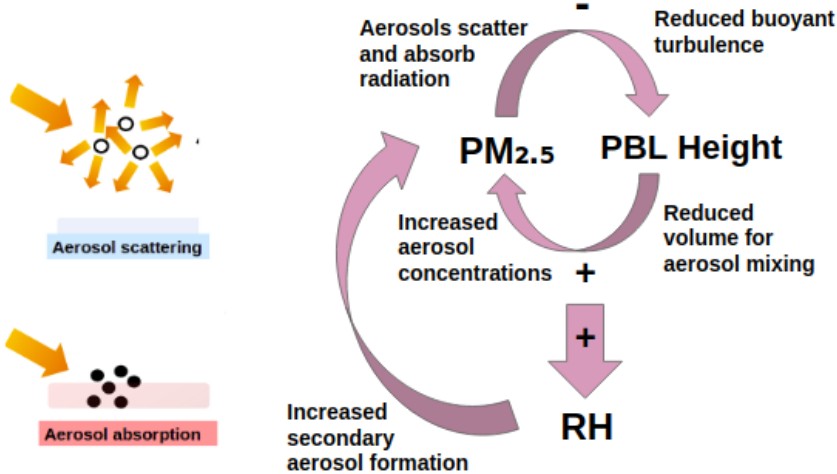

**Figure 1.** Schematic of the positive feedback loop between aerosols, radiation and meteorology thought to enhance pollution episodes in Beijing

     Aerosol composition and size are the main factors impacting an aerosol particle's single scattering albedo thus impacting the extent by which it will interact with radiation. Most aerosol particles predominately scatter radiation and thus have an overall cooling effect, stabilising the boundary layer and allowing for further accumulation of aerosol particles. However, black carbon (BC), an absorbing aerosol which can contribute up to 10 % of PM in Beijing (Liu et al., 2016) has the potential to have the
opposite effect, through warming of the lower atmosphere, which promotes buoyancy and destabilises the boundary layer.



However, depending on the vertical distribution of the BC layer, BC can also enhance stratification through causing warming in the upper PBL (Liu et al., 2018b; Zhong et al., 2018a; Ding et al., 2016).

Research examining the feedback effect on Beijing haze episodes has thus far relied upon observations or regional modelling studies. Liu et al. (2018b), Zhong et al. (2018b), Gao et al. (2015) and Wu et al. (2019) performed model simulations of pollution episodes using the Weather Research and Forecasting model with added chemistry (WRF-CHEM) to examine the feedback effect. Their results all confirm that aerosol-radiation interactions, aerosol hygroscopic growth and aqueous heterogeneous reactions all factor in the suppression of boundary layer development and result in increased surface $PM_{2.5}$ concentrations during polluted episodes in the North China Plain. Gao et al. (2015) suggests that aerosol-radiation interactions decrease temperature and shortwave (SW) radiation at the surface while increasing them aloft (925 hPa). Examining the feedback from a quantitative perspective, Wu et al. (2019) found that when $PM_{2.5}$ increased from 50 to 200 $\mu$g/m$^3$, maximum average boundary layer height decreased from 700 to 400 m. Furthermore, Zhong et al. (2019a) suggested that threshold $PM_{2.5}$ concentrations of 75 – 100 $\mu$g/m$^3$ exist in Beijing, above which the feedback effect is increasingly important and leads to aerosol accumulation and exacerbation of pollution episodes.

Observational studies also show a link between aerosol concentrations and boundary layer meteorology. Zou et al. (2017) studied the impact of high aerosol concentrations ($PM_{2.5} > 75$ $\mu$g/m$^3$) on Beijing meteorology over a year long period. Their results demonstrate that the aerosol impact on meteorology was different depending on the season, with particularly large reductions in sensible heat flux (SHF), PBL height and surface SW radiation reported in autumn and winter Liu et al. (2019) used the same $PM_{2.5}$ threshold to estimate the impact of high aerosol concentrations on observed meteorological data over a one month period where haze episodes occurred every 4-7 days. Comparing high and low aerosol periods they found that on average surface SW radiation was 36 % lower and daily maximum PBL height was reduced from 1.3 km to 0.6 km.

Despite an increase in research in this area, quantification of aerosol perturbations on boundary layer meteorology is still uncertain. In WRF-CHEM, results are strongly dependent on the boundary layer scheme or parameterisation employed throughout the simulations, while observations of this effect, although useful, only show links between the phenomena without being able to quantify the processes or separate factors. High resolution sensitivity studies which allow for direct analysis of boundary layer meteorology are therefore needed to be able to assimilate the major contributions to haze events.

Large-eddy simulations (LES) can explicitly resolve large, high energy eddies while parameterising smaller eddies for computational efficiency. This allows for direct investigation of boundary layer meteorology, turbulent fluxes and statistics, while easily controlled conditions allow for insight into the sensitivity of aerosol interactions on PBL dynamics (Liu et al., 2018b; Mazoyer et al., 2017). Several studies have used LES models to examine the impact of aerosols on convective boundary layers, cumulus clouds and radiation fogs, primarily in rural or marine environments (Mukherjee et al., 2016; Tonttila et al., 2017; Bellon and Stevens, 2012; Sullivan and Patton, 2011; Andrejczuk et al., 2014). In this work, a novel LES with a coupled sectional aerosol module (UCLALES-SALSA) has been developed to make it suitable for the urban environment of Beijing. The newly coupled aerosol-radiation scheme has been tested for the first time, in order to examine the feedback effect of



aerosol loading on boundary layer dynamics. Model description and details of set up for an urban environment are outlined in

75   section 2, section 3 describes the experimental set up for cases 1, 2 and 3, section 4 shows results of the simulations and section

5 discusses the results, including sensitivity of UCLALES-SALSA to: 5.1 - Meteorological conditions, 5.2 – Aerosol loading

and 5.3- Aerosol vertical profiles.

## 2   Model Description

The model used in this work is UCLALES-SALSA. A comprehensive description of the model and its previous uses can be

found in the paper by Tonttila et al. (2017). The version used here can be downloaded at https://www.github.com/UCLALES-

SALSA. A description of the model set up, validation and sensitivity to parameters are described below.

### 2.1   UCLALES

UCLALES is a large eddy simulation which has mainly been used in idealised cloud and fog studies. It is based on the

Smagorinsky–Lilly subgrid model and solves the Ogura–Phillips anelastic equations with an Asselin filter. Boundary conditions

are doubly periodic in the horizontal and fixed in the vertical. Momentum variables are advected with leapfrog time stepping

and scalar variables through forward time stepping. In the standard model a two-moment warm rain microphysical scheme is

used, the vertical is spanned by a stretchable grid and a sponge layer is applied at the domain top to prevent gravity waves

being released into the boundary (Stevens et al., 2003, 2005; Tonttila et al., 2017). The surface scheme explicitly calculates

sensible (SHF) and latent heat (LHF) fluxes at each time step and is based on a coupled soil moisture and surface temperature

scheme by Ács et al. (1991) (Eq.1, 2 and 3).

$$SHF = \rho C_p \left( \frac{(T_g - T_a)}{(r_a)} \right) \tag{1}$$

$$LHF = \frac{(\rho C_p)}{\gamma} \frac{(f_h e_s(T_g) - e_a)}{(r_{surf} + r_a)} \tag{2}$$

Where $\rho$ is air density, $C_p$ is specific heat capacity of dry air, $T_g$ and $T_a$ are surface and air temperature respectively, $\gamma$ is

the psychrometric constant, $f_h$ is a dimensionless function related to water volume fraction and takes the value 0.267 in our

case. $e_s(T_g)$ is saturation vapour pressure at surface temperature ($T_g$) and $e_s$ is water vapour at 2 m height. $r_{surf}$ is the surface

resistance to bare soil and is related to surface friction velocity (u*). $r_a$ is atmospheric resistance to water vapour and heat and

is dependent on atmospheric stability (Ács et al., 1991).

$$\Delta Q_s = \left( \frac{\omega C_h \lambda}{2} \right)^{\frac{1}{2}} (T_g - \bar{T}) \tag{3}$$





Surface parameters, which vary greatly in different environments, can be varied within the model input and largely affect the heat storage term ($\Delta Q_s$) (Eq.3). Where $C_h$ is volumetric heat capacity (J m$^{-3}K^{-1}$), $\lambda$ is thermal conductivity (W m$^{-1}$ K$^{-1}$), $\omega$ is angular frequency (s$^{-1}$) and $\bar{T}$ (K) is the average daily temperature in the 2 cm soil layer. The resulting parameters as well as the overall radiation are utilised in the surface energy balance scheme detailed in (Eq. 4) where Q$^*$ is net all wave radiation.

$$Q^* = H + LE + \Delta Q_s \tag{4}$$

### 2.2    SALSA

The Sectional Aerosol Scheme for Large Scale Applications (SALSA), was developed by Kokkola et al. (2008) and has been coupled with large eddy simulation models (UCLALES) as well as a climate model (ECHAM) (Kokkola et al., 2018; Tonttila et al., 2017). SALSA bins aerosols according to size, allowing for a variety of aerosol sizes and compositions as well as for aerosols to be either internally or externally mixed. (Kokkola et al., 2008) When SALSA is used in these simulations, aerosol

species included are black carbon, sulphate, organic carbon, nitrate and ammonium, with all aerosols assumed to be internally mixed. In terms of aerosol processes- coagulation and water vapour condensation are switched on, while nucleation, aerosol deposition, emissions and semi-volatile condensation are not considered here for simplicity but may be considered in future work.

### 2.3    UCLALES-SALSA

UCLALES-SALSA couples the UCLALES with SALSA and is primarily described in the paper by Tonttila et al. (2017). The version of UCLALES-SALSA here is a fully coupled radiation-dynamical model, whereby the aerosol-radiation interactions in SALSA are fully coupled with the four stream radiative solver in UCLALES which feeds back on boundary layer turbulence. This is the first time that aerosol-radiation interactions have been dynamically coupled to UCLALES and the work outlined here examines the sensitivity of aerosol loading on these interactions and feedback.

### 2.3.1    Aerosol-radiation interactions

The solution for radiative transfer in UCLALES is based on the 4-stream method integrating over 6 shortwave bands and 12 longwave bands according to Fu and Liou (1993). In this work, the scheme has been adapted to account for the sectional size distribution of the atmospheric aerosol. To this end we use pre-compiled look-up tables of the aerosol extinction cross-section, asymmetry parameter and single scattering albedo, which are given as a function of the size parameter (particle diameter

divided by wavelength) and the real and imaginary parts of the refractive index. For a given aerosol constituent, the refractive indices are catalogued at specific wavelengths. Nearest-neighbour interpolation is used to find the values closest to the centres of the wavelength bands used by the radiation solver. Assuming a perfect internal mixture of all aerosol constituents within one aerosol size section, the refractive index in that size section is then calculated as a volume-weighted average of its constituents.





This yields the optical thickness, single scattering albedo and phase function parameters weighted by the actual particle size
distribution resolved by SALSA, which are then taken to the 4-stream integration.(Fu and Liou, 1993)

### 2.3.2 Set up in an urban environment

In the past few decades, rapid urbanisation has transformed the landscape in Beijing, creating a microclimate which can be
represented by its own distinct physics. Part of this is the Urban Heat Island (UHI), which refers to the phenomenon where a city
is significantly warmer than its surrounding areas. This is a result of: increased SW radiation absorption, decreased longwave
(LW) radiation loss, decreased turbulent transport, increased heat storage and anthropogenic heat sources. Furthermore, urban
environments often consist of mainly impervious surfaces, and therefore the urban heat island is also often characterised by
low latent heat and comparatively higher sensible heat fluxes (Oke, 1982; Tong et al., 2017; Yang et al., 2016; Ikeda et al.,
2012). To set up UCLALES for an urban environment, alterations to the surface energy balance equation (3) were performed.

Studies by Oke (1982) outline two terms which can be used to represent the presence of the urban heat island. The first
is the alteration to $\Delta Q_s$ or the heat storage term which alters the rate of surface absorption and re-release of heat. In an
urban environment, typically the surface has higher surface heat capacity ($C_h$), water fraction, soil hygroscopicity and lower
thermal conductivity ($\lambda$) compared to rural environments. This subsequently feeds back on the surface temperature and heat
fluxes (Eq.4) The second term is an additional anthropogenic heat flux ($Q_f$), which accounts for all activities which result in
additional heat in a city. This can be split into heat from: buildings, industry, transport and human metabolism. Estimates of the
anthropogenic heat flux are difficult to perform and have not been done in wintertime Beijing, although a recent study gives
anthropogenic heat estimates for the summertime, which have a mean midday value of 67.2 W/m$^2$ (Dou et al., 2019). The
anthropogenic heat flux has a distinct diurnal profile, attuned to anthropogenic activities within a given city. It is high in the
daytime and decreases at night. The additional term is included in the surface energy balance scheme for an urban environment
as described in equation 5 (Grimmond and Oke, 1999; Hu et al., 2012; Schwarz et al., 2011; Xie et al., 2016; Yang et al., 2016).

$$Q^* + Q_f = SHF + LHF + \Delta Q_s \tag{5}$$


In order to set up UCLALES-SALSA for an urban environment, alterations to the heat storage term and a simplistic addi-
tional anthropogenic heat flux were included in the surface scheme and sensitivity studies were performed for a non polluted
day in Beijing (Figure 2) Simulation results were compared with observations taken during the Air Pollution and Human Health
(APHH) Beijing field campaign as well as with ECMWF and radiosonde meteorological profiles. The 22nd November 2016
was chosen for the initial sensitivity simulations. As a non polluted day in Beijing, observations on 22nd November are not im-
pacted by aerosol interactions. Potential temperature, moisture and wind profiles were taken from ECMWF ERA-5 reanalysis
data and surface meteorological values taken from an automatic weather station based at the Institute for Atmospheric Physics
(IAP) in Beijing. In the first simulation, with no adaptation to the surface scheme there was a clear discrepancy between mod-
elled and measured sensible and latent heat flux and potential temperature profiles. Particularly, there was a large difference in





the lower potential temperature profiles in the evening, where the modelled simulations showed early radiative cooling when

compared to observations. Delayed and reduced radiative cooling at the surface is frequently observed in urban environments

including Beijing.

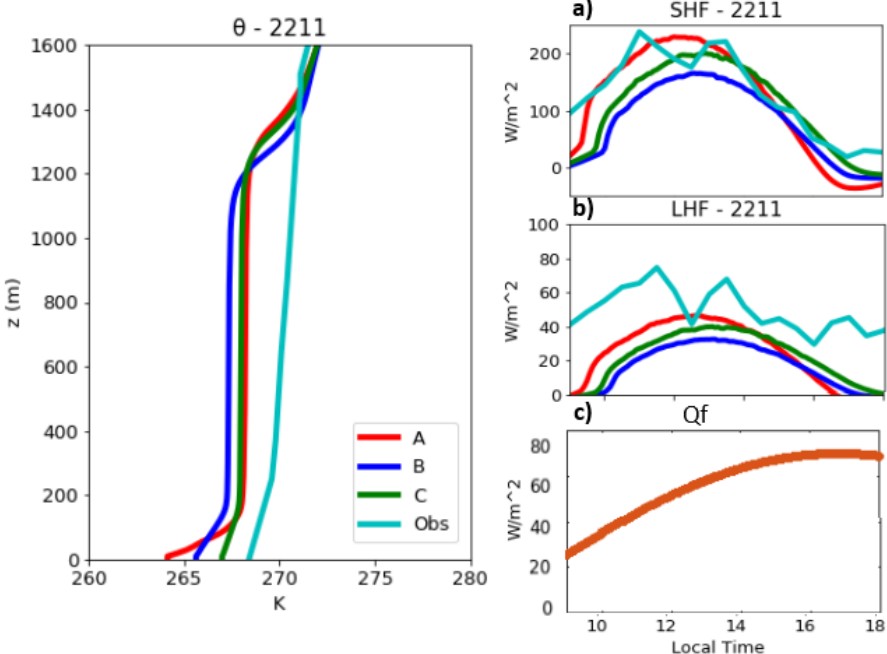

**Figure 2.** Potential temperature ($\theta$) profiles at 8 pm (left), a) Sensible Heat Flux (SHF) and b) Latent Heat Flux (LE) diurnal profiles (right)
for no anthropogenic heat flux (A (Red) – $C_h = 2\times10^6$ (J m$^{-3}$K$^{-}$1), B (Blue) – $C_h = 7\times10^6$) and an anthropogenic heat flux (C (Green) –
$C_h = 7\times10^6$) and observations. Where c) shows anthropogenic heat flux ($Q_f$) used in the simulation

Of all surface parameters altered, the largest sensitivity the model showed was to volumetric heat capacity ($C_h$). Increas-

ing this term decreased maximum SHF, noticeably delayed nocturnal radiative cooling and slightly lowered the temperature

through the profile (Figure 2). This is due to slower release of outgoing radiation, which is stored for longer in urban surfaces.

Figure 2 shows the sensitivity to varying surface volumetric heat capacity (J m$^{-3}$K$^{-1}$) between the initial value ($2\times10^6$) and

chosen value ($7\times10^6$). Higher volumetric heat capacity of the surface causes delayed nocturnal cooling, resulting in higher

sensible and latent heat flux in the evening. The surface urban energy balance is also affected by an anthropogenic heat flux

which varies seasonally and spatially. A diurnal anthropogenic heat flux which peaks at 70 W/m$^2$ during the daytime and re-

mains around 20 W/m$^2$ in the evening was included in a further simulation. Inclusion of a diurnal $Q_f$ profile increased overall

temperatures as well as latent and sensible heat fluxes (Figure 2).

This sensitivity work provides the setup for UCLALES-SALSA in an urban environment and this is utilised for the remainder

of results presented below which all include a diurnal $Q_f$ profile and heat capacity ($C_h$) set at $7\times10^6$ Jm$^{-3}$K$^{-}$1 , which is a





value typical of concrete (Takebayashi and Moriyama, 2012). The scope for variation of surface parameters within UCLALES

is extremely high, therefore we recognise that within the model framework there is a strong dependence on parameters such as temperature, roughness, heat capacity, albedo and soil moisture. It is also likely that due to the simple homogeneous surface scheme used, some features of the urban environment that are observed cannot be replicated in the chosen model framework. Although the effect of these features is important to understand, the purpose of this paper is to examine the suitability of using an LES model in investigating urban haze. The parameters chosen here are based on identification of the urban measurement

site's characteristics, as well as from chosen literature values and are to the best of the authors' knowledge a fair representation of urban Beijing, as described in the next section.

## 3  Experimental Method

### 3.1  Observational Data

All measurements used in this study were taken at the Institute of Atmospheric Physics (IAP), Chinese Academy of Sciences,

as part of the APHH Beijing campaign. Measurements taken include but are not limited to: NR-PM$_1$ (non refractory PM with a diameter < 1 $\mu$m) composition and aerosol and black carbon size and concentration measurements at the surface, as well as meteorological measurements at 15 levels on a 320 m tower. Sensible and latent heat flux measurements were calculated and a ceilometer was used to infer PBL height. For more information concerning the measurements taken as well as the APHH project and field campaign the reader is directed to the 'Introduction to special issue (APHH Beijing)' by Shi et al. (2018) .

### 3.2  Experimental setup

The domain size for all model simulations spanned 5.4 km in the horizontal, with a resolution of 30 m and the model top was set to 1.8 km in the vertical with a resolution of 10 m. The model uses an adaptive timestep with a maximum timestep of 1 s. A haze period which took place within the APHH winter campaign period from 24th - 26th November 2016 was used to examine the sensitivity of boundary layer meteorology to varying aerosol concentrations. Meteorological data taken

from ECMWF-ERA5 reanalysis and tower meteorological data was used to initialise vertical profiles at 8am (local time) on 24/11, 25/11 and 26/11. Simulations were run for 14 hours (10pm) including 1 hour spin up time. Simulations for all days were considered to be cloudless. Case studies for each day were simulated and compared to each other and are described as follows: Case 1 – No aerosols, Case 2-High and low aerosol loading, Case 3-Aerosol vertical profiles. For case 2 aerosol vertical profiles were constant in the column whereas case 3 examined the impact of including a varying aerosol vertical profile.

Aerosol size distribution parameters and volume fraction of aerosol components were the same for all simulations, detailed in tables 1 and 2. In all cases, BC can be considered to be the primary absorbing aerosol, with SO$_4$ strongly scattering and OC predominantly scattering with a small absorbing component, while both NO$_3$ and NH$_4$ do not directly interact with radiation in these cases Aerosol growth is considered through coagulation and condensation of water vapour but for simplicity semi-volatile condensation and dry deposition is switched off and no additional emissions are considered.





|  | Low | High |
|---|---|---|
| $D_g$ (nm) | 100 | 100 |
| $\sigma_g$ | 1.55 | 1.55 |
| N (#/mg) | 10,000 | 40,000 |
| PM ($\mu g/m^3$) | 55 | 220 |

**Table 1.** Size distribution parameters initialised for simulations examining measured aerosol feedback on meteorology. Dg (geometric mean diameter), $\sigma g$ (geometric standard deviation), N (number concentration), as well as calculated surface PM concentration for low and high aerosol simulations

| Composition | Fraction |
|---|---|
| OC | 0.5 |
| $SO_4$ | 0.1 |
| $NO_3$ | 0.21 |
| $NH_4$ | 0.09 |
| BC | 0.1 |

**Table 2.** Volume fraction of aerosols included in SALSA for all simulations in case 2 and case 3


## 4  Results

The results highlighted in this section aim to test the sensitivity of the newly coupled aerosol-radiation scheme in UCLALES-SALSA to aerosol loading, using meteorological conditions, urban characteristics and simplified aerosol conditions, associated with Beijing haze episodes. Case 1 shows boundary layer development for 24/11, 25/11 and 26/11 with no aerosols, case 2
examines the effect of high and low aerosol loading for each of the days and case 3 focuses on the impact of varying aerosol vertical profiles.

### 4.1  Case 1- No Aerosols

Simulations in case 1 examine the development of boundary layer dynamics for 24/11, 25/11 and 26/11, without aerosol-radiation interactions. All 3 days are initialised with different meteorological vertical profiles, taken from ECMWF profiles.
On 25/11 there is a strong temperature inversion throughout the whole profile, while on 26/11 there is strong vertical wind shear, higher surface humidity and strong stability in the lowest 300 m (Figure 3). Strong vertical wind shear causes mechanical turbulence, while a strong temperature inversion in the morning can suppress boundary layer development through reducing





buoyancy. Figure 3 (right) shows development of SHF, PBL height and total turbulent kinetic energy (TKE) for the three simulated days with different meteorological conditions initialised in the morning.

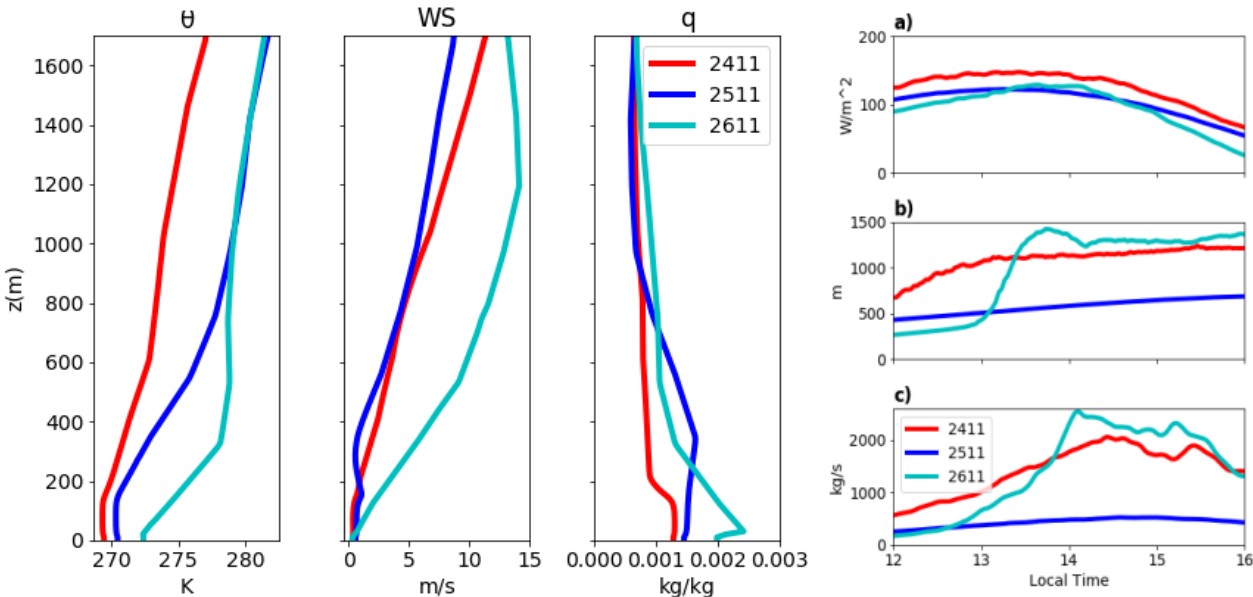

**Figure 3.** Left - Initial vertical profiles of Potential Temperature ($\theta$), Wind Speed (WS) and Total Water Mixing Ratio (q) and Right - a) Sensible Heat Flux, b) Height of maximum gradient in theta, c) Vertical integral of TKE for 24/11 (red), 25/11 (blue) and 26/11 (turquoise) for simulations with no aerosols (Case 1)

SHF is similar in magnitude for all 3 days, while TKE and simulated PBL height is significantly lower for the 25/11 simulation. A well mixed, turbulent boundary layer forms quickly on 24/11, however, on 25/11 a shallow, weakly turbulent boundary layer remains throughout the day and on 26/11 a turbulent boundary layer is much slower to develop (Figure 3). The changing conditions used here are typical for a Beijing haze episode and show that even without the consideration of aerosols, meteorological conditions can largely affect the diurnal development of boundary layer dynamics.

**4.2   Case 2- High and Low Aerosol Loading**

Case 1 shows that simulated boundary layer dynamics are impacted by initial meteorological conditions. In case 2, the sensitivity of boundary layer dynamics to aerosol loading is examined, where aerosol mixing ratios were constant throughout the profile (Figure 5). Table 3 shows the impact of including high and low aerosol loading on maximum SHF and maximum PBL height between 12:00 and 16:00 LST (Local Standard Time).

Potential temperature profiles show that inclusion of aerosols causes cooling in the lower boundary layer and warming in the upper layers as well as reducing overall PBL height (Figure 4), while also reducing surface wind speed (Figure 5) Including





high aerosol concentrations (220 $\mu g/m^3$) compared to low aerosol concentrations (55 $\mu g/m^3$) maximised this effect, causing a significant reduction in SHF and PBL height on all 3 days (Table 3)

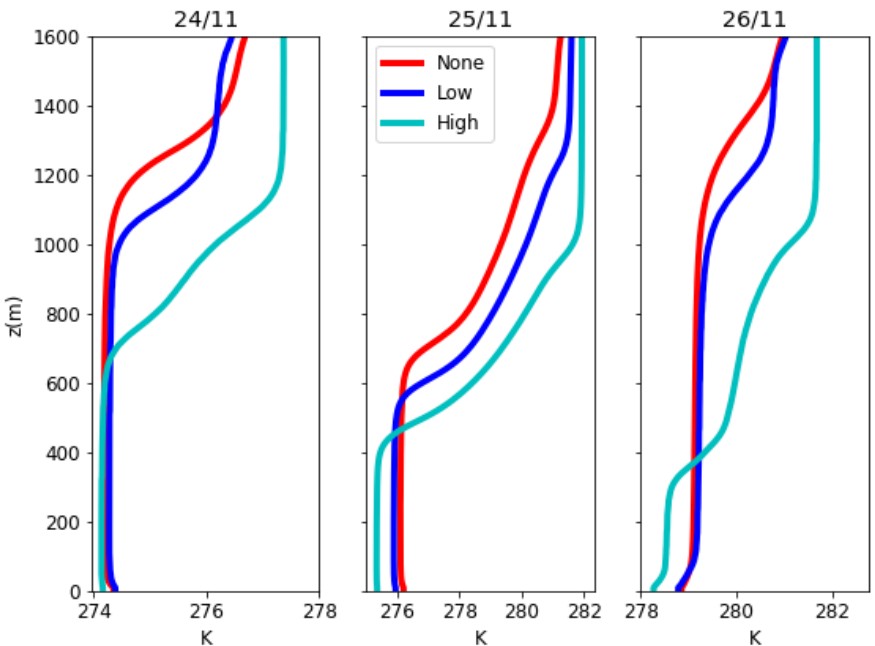

**Figure 4.** Potential temperature profiles at 5pm for 24/11, 25/11 and 26/11, with no aerosols (red), low aerosol loading (blue) and high aerosol loading(turquoise)

| Day | 24/11 | 25/11 | 26/11 |
|---|---|---|---|
| Max PBL height (None) | 1240 | 695 | 1424 |
| Max PBL height (Low) | 1088 | 592 | 1169 |
| Max PBL height (High) | 915 | 475 | 391 |
| % decrease in PBL height (High-Low) | 16% | 20% | 67% |
| Max SHF (None) | 148 | 123 | 129 |
| Max SHF (Low) | 126 | 97 | 100 |
| Max SHF (High) | 82 | 64 | 55 |
| % Decrease in SHF (High-Low) | 35% | 34% | 45% |

**Table 3.** Change in maximum PBL height (taken as the height between 12 and 4pm with a maximum gradient in potential temperature) and SHF for all 3 days with high and low aerosol loading

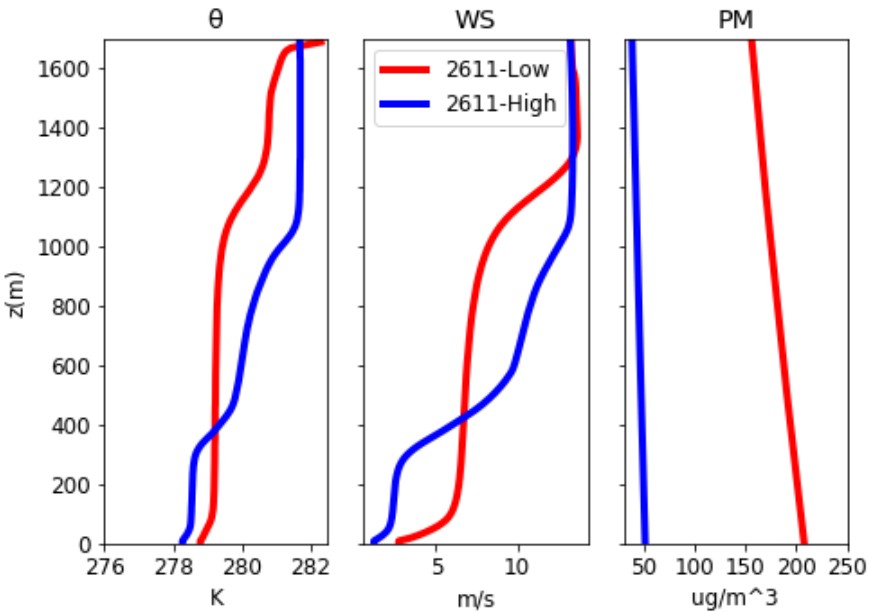

**Figure 5.** Potential Temperature ($\theta$), Wind Speed (WS) and Aerosol Concentration (PM) profiles on 26/11 for high (red) and low (blue) aerosol concentrations at 5pm (9 hours of simulation)

### 4.3 Case 3- Aerosol Vertical Profiles

To assess the sensitivity of the model to a varied aerosol vertical profile, case 3 uses the same set up as case 2 but varies the aerosol mass mixing ratio with altitude, as shown in figure 6. This is to assess the impact of high aerosol concentrations aloft in case 2 simulations which may magnify the aerosol-radiation effect, due to higher total loading increasing the total column aerosol optical depth (AOD). In case 3 simulations, total aerosol mass loading throughout the column is $\sim$ 22 % less than for case 2 simulations for both high and low aerosol simulations. The aerosol profile was chosen so that at the first time step, aerosol mass mixing ratio at the surface was the same as those with a constant profile and decreased above the PBL in accordance with the potential temperature profiles, while composition and size remained constant throughout (Figure 6).

Figure 7 shows simulation results of potential temperature and aerosol number mixing ratio at 5pm (9 hours of simulation) for constant and varied aerosol vertical profiles at high concentrations for 24/11 and 26/11. When a varied aerosol profile is included, vertical mixing of aerosol occurs, resulting in a difference in the aerosol vertical profile on each day at 5pm due to the difference in meteorology. The difference between the aerosol profiles over time shows the modelled meteorological feedback on aerosol mixing ratios.





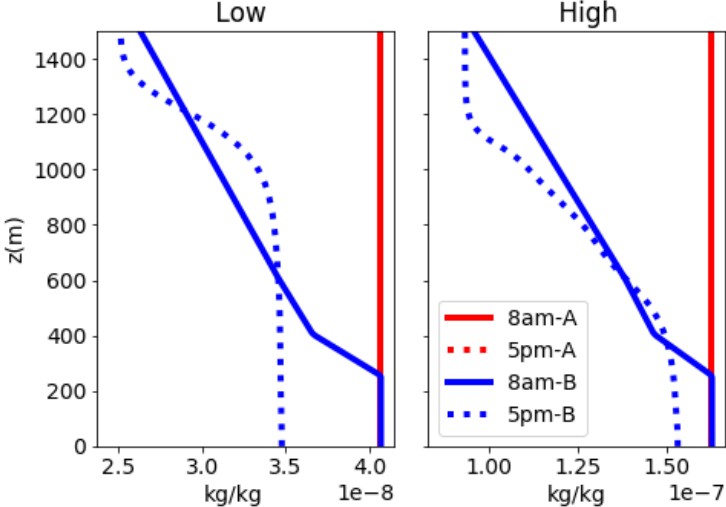

**Figure 6.** Aerosol mass mixing ratio vertical profiles for low and high aerosol loading simulations on 26/11, for constant aerosol profile (red) and vertically varying aerosol profiles (blue) at initial timestep (solid) and after 9 hours simulation (dashed)

Figure 8 compares the variance in vertical velocity ($\sigma_w^2$) for low and high aerosol loading throughout the profile and at high aerosol loading at the surface only for 24/11 and 26/11 simulation. Showing that high aerosol loading both throughout the

column and at the surface, decreases $\sigma_w^2$ at 500 - 750 m in the afternoon of both 24/11 and 26/11. It also shows the effect of high aerosol loading throughout the column, which causes an increase in $\sigma_w^2$ close to the model top, with the varied aerosol vertical profiles minimising this effect.

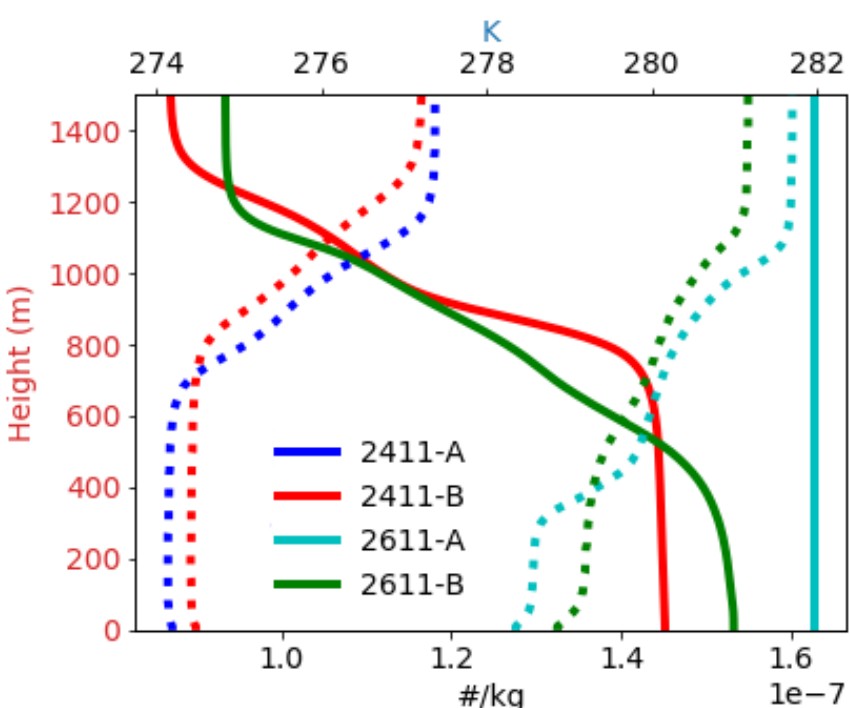

**Figure 7.** Number mixing ratio (solid) and potential temperature (dashed) vertical profiles at 5pm for constant vertical aerosol profiles on 24/11 (Blue) and 26/11 (Turquoise) and varied aerosol vertical profiles on 24/11 (Red) and 26/11 (Green)





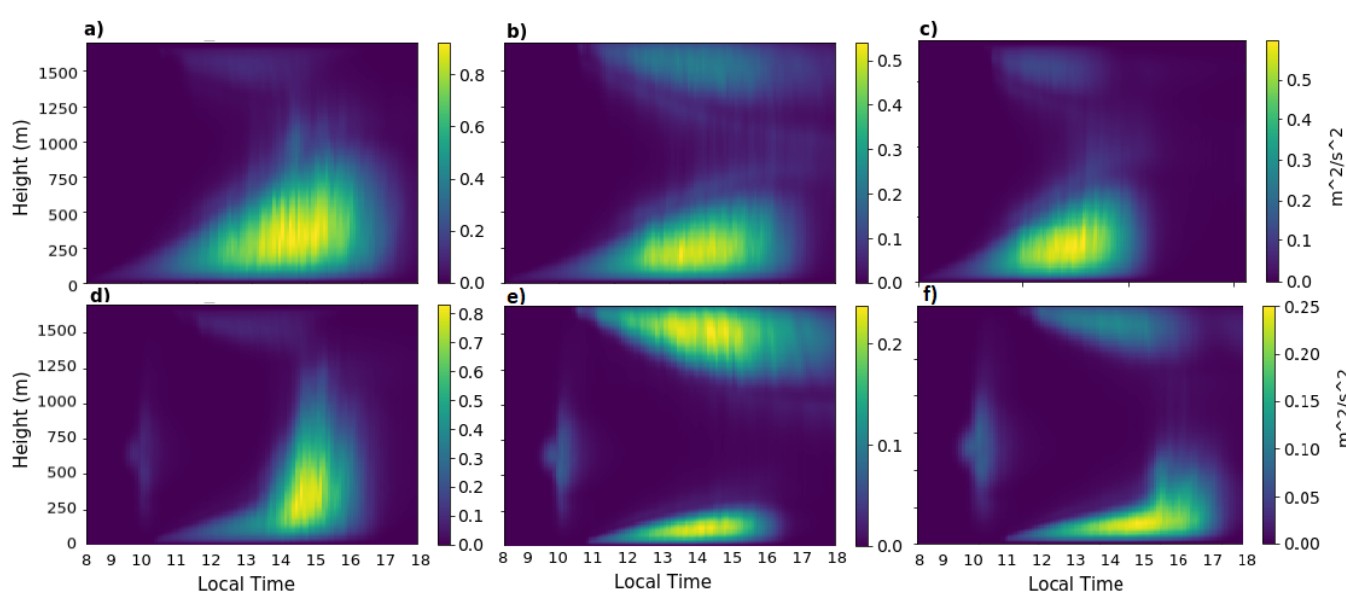

**Figure 8.** Variance in vertical velocity ($\sigma_w^2$) for case 2 and case 3 simulations on 24/11 (a-c) and 26/11 (d-f) for low aerosol loading (a and d), high aerosol loading (b and e) and high loading with a varied aerosol vertical profile (c and f)



## 5 Discussions

The results highlighted above show the use of a novel coupled LES-aerosol radiation model to investigate haze in the urban
environment of Beijing. Simulated sensitivity to urban surface parameters is high and these will be different for other urban
locations. It is therefore necessary to evaluate and tune these parameters to observations in specific environments in order to
use an LES model to fully explore boundary layer dynamic sensitivities. Aerosol-radiation interactions were tested for the first
time in the model framework and showed that sensitivity of boundary layer meteorology and turbulence to aerosol loading was
strong while also being dependent on initial meteorological conditions.

### 5.1 Sensitivity to meteorology

Case 1 identifies the importance of meteorological conditions on boundary layer dynamics throughout the day. Many obser-
vations in Beijing found that meteorological conditions are a main driver on both the onset and longevity of haze. Large scale
synoptic conditions such as southerly winds and low pressure often preempt pollution episodes which tend to occur every 4-7
days in Beijing wintertime (Liu et al., 2018a; Wang et al., 2019). These conditions are associated with the beginning of 'haze'
as the switch in meteorological conditions from strong northwesterly to southerly winds advects pollution from surrounding
provinces into Beijing. This change is also associated with a low pressure field within the city, where stagnant air becomes
trapped and the dispersion of pollutants is inhibited (Gao et al., 2016).

The initial meteorological profiles for the simulations on 24/11 are taken prior to the onset of the haze and are associated
with clean conditions. This is likely the reason for the quick turbulent boundary layer development along with high TKE
and SHF throughout the day. Observations show that aerosol concentrations begin to build up around midday on 24/11 and
remain constant until the afternoon of 25/11 when concentrations build up rapidly, peaking overnight on 25/11 and remaining
high until the afternoon of 26/11. Therefore, the initial conditions used in the simulation of 25/11 will have been affected by
aerosol-radiation interactions of the previous evening.This explains the strong temperature inversion in the morning and results
in a shallow turbulent boundary layer forming in these simulations, with lower turbulent kinetic energy compared to the 24/11
simulation (Figure 3).

### 5.2 Sensitivity to aerosol loading

Aerosol-radiation interactions cause a reduction in SHF, surface SW radiation and TKE resulting in a reduction in the daily
maximum PBL height for all three days examined. High aerosol concentrations enhance this effect due to an increased number
of aerosols interacting with radiation.This leads to a reduction in maximum SHF of 44, 33 and 45 W/m$^2$ for high compared to
low aerosol loading simulations on 24/11, 25/11 and 26/11 respectively. However, results from case 2 show a variation in the
magnitude of the aerosol-radiation effect with a larger impact on maximum PBL height for high aerosol simulations on 26/11
compared to 24/11 and 25/11 (Table 3). Including high aerosols on 26/11 causes $> 1$ $^o$C of daytime cooling in the lowest 300m
compared to 0.3 $^o$C of cooling on 24/11 (Figure 6). The larger degree of cooling on 26/11 leads to a larger reduction in buoyant
turbulence and prevents the full growth of a deeply turbulent boundary layer to a larger extent on 26/11.


High aerosol concentrations are known to stabilise the boundary layer through the reduction of vertical transport of momentum to the surface (Jacobson and Kaufman, 2006). This can reduce wind speeds at lower altitudes and thus decrease wind shear and the shear component of TKE. In case 2, high aerosol loadings reduce surface wind speeds, wind shear and surface frictional velocity (u*) for all 3 days, with a greater reduction on 26/11 compared to 24/11 and 25/11. High aerosol loading also causes a reduction in the variance of vertical velocity ($\sigma_w^2$), which can be considered a measure of turbulence (Stull, 2015). On

both 24/11 and 26/11, simulations with high aerosol loading caused a reduction in the magnitude of $\sigma_w^2$ particularly between 500-800 m. On 24/11, the decrease at the surface at $\sigma_w^2$ is $\sim 40$ %, while on 26/11 the reduction is 75 %. In the case of 26/11, this is accompanied by increased values of $\sigma_w^2$ in the upper layers close to model top, which results in two turbulent layers forming separated by a stable layer. (Figure 8).

        In these simulations, aerosol profiles are constant through the column and high aerosol concentrations aloft Figure 7 shows

that high aerosol throughout the column causes warming in the upper layers and cooling in the lower layers, which causes strong stability throughout the profile. In reality, aerosols tend to be concentrated closer to the surface and within the boundary layer, although occasionally in Beijing regional transport can lead to higher aerosol concentrations aloft. Therefore, case 3 investigated the effect of limiting pollution to the surface by including aerosol vertical varying profiles.

## 5.3   Vertical profiles

Case 3 examined the impact of meteorological feedback on aerosol vertical mixing for high and low aerosol loading simulations by including aerosol vertical profiles on 24/11 and 26/11. Simulations with a varied vertical aerosol profile had the same aerosol concentrations at the surface but reduced concentrations at higher altitudes (Figure 6). This resulted in a small increase in maximum SHF ($\sim 7$ W/m$^2$) on both 24/11 and 26/11. For 26/11, limiting high aerosol loading to the surface results in an afternoon increase in turbulence up to 500 m. Furthermore, The effect of high aerosols throughout the column (case 2) resulted

in a highly turbulent layer at model top and a large reduction in surface wind speed on 26/11 (Figure 8). As this turbulent layer is significantly reduced with lower aerosols aloft, this effect may be considered to be an artefact of aerosol loading at high altitudes which is not often observed in poor air quality events during wintertime in China. However, overall the contribution of the shear term to turbulence is minimal compared to the buoyancy term, which is greatly reduced by high aerosol loading in both case 2 and case 3. The high aerosol loading in case 2 has a larger effect on boundary layer development than the effect

of a varying the aerosol vertical profile in case 3. Therefore, we can consider that the change in the thermal profile of the atmosphere, due to high concentrations of aerosols increasing aerosol-radiation interactions, to be the prominent cause of the reduction in SHF and PBL height (Table 3).

        The large degree of cooling on 26/11 compared to 24/11 is due to the effects of initial meteorology feeding back on aerosol-radiation interactions. Figure 7 shows potential temperature and aerosol number mixing ratio vertical profiles for each case

(after 9 hours of simulation) under high aerosol loading at a) the surface only and b) throughout the profile. After 9 hours of simulation (5pm LST) surface aerosol concentrations on 26/11 are higher than on 24/11. This is due to aerosol-radiation interactions and initial meteorological conditions on 26/11 resulting in a shallower PBL (Table 3). This shows the ability of





UCLALES-SALSA to simulate the aerosol-radiation-meteorological feedback loop and that the feedback effect can have a significant impact on aerosol surface concentrations, which will consequently feedback further on atmospheric stability.

## 6  Conclusions

UCLALES-SALSA was set up to model an urban environment for the first time, in order to investigate the impact of aerosol-radiation interactions on urban haze. During set up, sensitivity to urban surface parameters was shown to be high, and accounted for the slower release of heat throughout the day as observed in urban Beijing. Inclusion of a diurnal anthropogenic heat flux in simulations resulted in a warmer environment typical of an urban heat island. Given the sensitivity to such parameters, accurate measurements of these properties can be considered paramount in order to improve modelling of the urban environment. Turbulent motion throughout the day in each simulation is further impacted by initial meteorological profiles. Conditions associated with clean periods in Beijing allow for the development of a highly turbulent boundary layer, while strong morning temperature inversions prevent the growth of a turbulent boundary layer throughout the day. Aerosol-radiation interactions in all cases decreases SHF, TKE and PBL height, as well as causing cooling at the surface and reducing surface wind speeds. All simulations also show large sensitivity to aerosol loading, with more than a third reduction in SHF due to high aerosol loading in all simulations. Through comparing simulations with and without aerosol vertical profiles (case 3) we observe that on 26/11 the simulated development of a turbulent boundary layer in the afternoon is impacted by high aerosol loading aloft (case 2) This is due to aerosols at high altitudes reducing mechanical shear as well as the reduction in buoyancy. However, overall the effect of including a vertical aerosol profile is minimal compared to the effect of overall aerosol loading which suggests a higher effect of surface aerosols.

The sensitivity work outlined above aims to isolate the aerosol and dynamical effects on pollution episodes through using a specific period with varying meteorological conditions and simplified aerosol conditions. LES models are limited in their ability to represent changing synoptic conditions without additionally forcing or nudging simulated profiles with mesoscale model results or through observations. However, these simulations do show the sensitivity to and importance of meteorological conditions on the development of boundary layer turbulence in Beijing. As well as assessing the importance of aerosol loading on the aerosol-meteorology feedback loop and the impact on PBL turbulent statistics. The aerosol feedback loop is thought to have the largest impact on haze episodes during the cumulative and dissipation stages of the pollution episode. Future work will focus particularly on these stages and the impact of aerosol-radiation-meteorology interactions. As aerosol optical properties play an important role in the feedback, future work will also take advantage of the SALSA framework to vary aerosol optical properties in a case study of Beijing haze.



*Code availability.* The code used in this manuscript can be downloaded at https://www.github.com/UCLALES-SALSA

*Author contributions.* The idea for the study was conceived by JS, GM and HC. All model simulations were performed by JS with the assistance of JT. JS wrote the paper with input from JT and TK. All co-authors discussed the results and commented on the manuscript. The authors declare that they have no conflicts of interest.

350  *Competing interests.* The authors declare that they have no conflict of interest.

*Acknowledgements.* Jessica Slater is fully funded by the National Centre for Atmospheric Science (NCAS). This work was carried out as part of AIRPRO (NE/N00695X/1) for which Jessica Slater, Hugh Coe and Gordon McFiggans also acknowledge funding. Sami Romakkaniemi and Juha Tonttila are supported by the Academy of Finland (projects 283031 and 309127) Model simulations were carried out on the ARCHER UK National Supercomputing Service (http://www.archer.ac.uk). We gratefully acknowledge Yele Sun's group and Pingqing Fu's

355  group at IAP for aerosol composition data and tower meteorological data respectively, as well as Zifa Wang's group at Peking Univeristy for aerosol size data and Eiko Nemitz at CEH Edinburgh for heat flux data.



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
