# Peer review of "Using a coupled LES-aerosol radiation model to investigate urban haze: Sensitivity to aerosol loading and meteorological conditions"

_Atmospheric Chemistry and Physics, 2020_

## Referee Comment (RC1) · Anonymous Referee #1 · 25 Apr 2020

Aerosol-radiation interaction has been indicated to play a crucial role in boundary layer meteorology as well as near-surface air pollution by both observational and numerical works recently. This work used a coupled LES-aerosol radiation model to investigate the impact from aerosol loading and meteorological conditions on turbulent kinetic energy and planetary boundary layer evolution. One strength of this work is the coupling of aerosol module and LES model, rather than PBL parameterization, which can better resolve the PBL process, particularly under stagnant condition. Inclusion of aerosol/chemistry in LES model did improve our understanding in aerosol-boundary layer interaction. Overall, this work is well structured and written. Here are some issues that are suggested to be addressed for further improving this work.

[Figure]

In experimental setup, please specify how the aerosol size distribution and volume fraction of chemical composition were determined. Were they adopted by in-situ measurements? In addition, why not consider the impact of nitrate and ammonium on radiation transfer given that both of them contribute a large fraction of aerosol mass loading in Beijing. Also, a description of aerosol size bins should be necessary. Like how many bins are separated and what are their ranges?

The simulation was conducted for only three days. Why chose those days, were pollution or meteorology on 24-26 NOV. in 2016 very typical in Beijing during cold season?

Line 18: "aerodynamic diameter " is more precise here

Line 138: It seems like that the surface energy balance is represented by equation (4) rather than (3)?

Line 198: The constant concentration in the column may not be an appropriate representation of urban pollution since most emissions in the city located at surface (i.e., traffic). Especially the model is initialized at 8am in the morning when most pollutants are confined in a shallow near-surface layer.

Line 203-204:This sentence needs to be rephrased.

Figure 5: The legend is reversed for "high" (should be red) and "low" (should be blue).

Figure 6: Under the high aerosol loading situation, the entire layer of aerosol mass mixing ratio seems to decrease. Excluding the dry deposition why the column aerosol loading experienced a reduction?
* * *

---

## Referee Comment (RC2) · Anonymous Referee #2 · 7 May 2020

This study set up a coupled LES-aerosol radiation model to simulate an urban environment and investigate the interaction between aerosols and PBL dynamics. The coupled LES-aerosol model made it possible to get more information for the sensitivity of PBL dynamics to aerosols. This paper was well written, especially with a very informative description of the novel coupled model. The model results showed its ability to simulate the interaction between aerosol, meteorological condition, and PBL dynamics. However, I would suggest adding more discussion on the processes and mechanisms of the interaction between aerosol, meteorological condition, and PBL dynamics using this novel model. I recommend further expand the discussion section to help us understand deeper in the effect of aerosol on the PBL and the effect of interaction on the

development of haze episode.

Specific comments:

1. Why did you only simulate daytime but not the full day including night?

2. Figure 2(b): Why did all cases underestimate the latent heat flux to a large extent?

3. Line 203: Why did you assume SO4 strongly scattering but ignore NO3? As measured in SALSA, nitrate is much more than sulfate during this haze episode (in a factor of ~2), which is made up 21% of aerosols as you list in Table 2. I don't think the effect of NO3 can be ignored in this case. As well as NH4, which is as important as SO4 for the fraction of aerosols. As a result, your model result may underestimate the scattering of all aerosols.

4. Line 230: Although it is clear that aerosols have an effect on the decrease in the PBL height, the inclusion of high and low aerosol concentration has a different effect on the temperature in the lower and upper boundary layer as shown in Figure 4. For example, a low concentration of aerosol causes cooling in the upper layer, while the high concentration of aerosol causes warming there on 24/11; low concentration of aerosol causes slight cooling in the lower layer on 25/11, while cause slight warming on 26/11. I suggest adding some more detailed discussion instead of giving a rough conclusion as "inclusion of aerosols causes cooling in the lower boundary layer and warming in the upper layers".

5. Table 3: Could you add some discussion on the difference in the percentage of decrease in PBL height between high and low aerosol cases? The high concentration of aerosol causes a much larger decrease in PBL height on 26/11 than that on 24/11 and 25/11. This discussion would help to understand more about the effect of aerosol on PBL height and its influence factors. I saw you gave some explanation in Section 5.2, but I think those were still not enough. First, why high aerosols will cause >1°C cooling on 26/11, while only 0.3°C on 24/11? Second, it looks like the high aerosols

decrease the potential temperature $\sim1$°C in the lower layers on 25/11, but why PBL height was only decreased by 20%, which was much smaller than that on 26/11.

6. Line 252: Varied aerosol vertical profiles minimize the increase in vertical velocity close to the model top, while they also increase the vertical velocity in the lower layers. This section (4.3) was talking about the effect of aerosol vertical profiles, but this paragraph only mentioned it with a half-sentence. The comparison in the vertical velocity is interesting and informative as shown in Figure 8. Please give more results about the effect of aerosol vertical profiles. Maybe separate the effects of high aerosol concentration and aerosol vertical profiles on vertical velocity into two sections.

7. Line 273: Could you state more detail on how aerosol-radiation interactions of the previous night affect the meteorological condition on 25/11? Else, I suggest discussing more the sensitivity of boundary layer dynamics to meteorological conditions and its detail processes, instead of talking much about the variation of observed aerosols during these days.

8. Figure 7: Does the turquoise line mean the aerosol profile does not change (significantly) in the case with the constant aerosol profile? If yes, why not?

9. Line 317: This result was enough to prove the ability of your model to simulate the interaction between aerosol and meteorological conditions, but I would recommend adding more discussion on the detailed process of the interaction between aerosol and meteorological condition within the PBL using your model.

10. In this study, you mixed all kinds of aerosols and showed comprehensive results. Since the absorbing aerosol and scattering aerosol have a totally different effect on PBL dynamics, I would suggest conducting more sensitivity experiments to investigate the separate effect of absorbing aerosol and scattering aerosol. This would further improve our understanding of the interaction of aerosol and PBL during haze.

Technical comments:

1. Line 23: It will be better to add some references for these observation results.

2. Line 32: Change distribution to diffusion?

3. Line 125: what are the refractive indexes you used in the model? Please add a table or reference.

4. Line 158: What is "the first simulation"? Case A?

5. Figure 2: The potential temperatures at 8 pm are shown on the left, but you did not simulate anything at 8 pm. As you stated, the model ends at 6 pm each day. Why did you show the temperature at 8 pm here?

6. Line 178: What are some features you mentioned here? Please clarify them.

7. Line 201-202: SO4, NO3, NH4 are not very correct to stand for sulfate, nitrate, and ammonium.

8. Line 203: A period was missing after "these case".

9. Line 228: It is better to mention figures in order. Figure 5 was discussed in figure 4.

10. Figure 5: The color in the legend is wrong.

11. Figure 5: Why did not use a unit of mixing ratio in Figure 5 like Figure 6&7? Making them the same will be better to understand.

12. Figure 6: where are dashed red lines? Could you make them more obvious? Or give some explanation in the legend.

13. Figure 6&7: I suggest to change A and B to case 2 and case 3.

14. Figure 7: where is the blue solid line?

15. Figure 8: I suggest adding a short subtitle (like date & case) for each subplot to make readers easy to understand.

16. Line 284: Did you mean Figure 4?
17. Line 294: A period was missing after "aloft".

18. Line 303: same aerosol as high concentration case?

19. Line 315: What do a) and b) stand for here? I didn't find a subplot in Figure 7.

---

## Referee Comment (RC3) · Anonymous Referee #3 · 8 May 2020

General comments

Coarse-resolution models have been used to investigate the aerosol-radiation-meteorological feedback. The authors have developed the coupled LES-aerosol radiation model to investigate the sensitivity to aerosol loading and meteorological conditions. The results presented in this paper are interesting and the readability of the main text is excellent. I have some minor comments and questions to improve this paper.

Specific comments

p.4, l.83: Please define cloud and fog in the model. How did you consider the interaction with aerosols?

[Figure]

p.5, l.127: How did you consider the lens effect?

p.8, l.196: How did you set up the cloudless with varying aerosol concentrations?

p.8, l.201: Please specify the small absorbing component of OC.

p.8, l.203: How did you consider the gravitational setting of aerosols? Please explain the boundary condition for aerosols.

p.12, l.236, Figure 6: Why don't you show the mass concentration as in Figure 5? You do not have to repeat the constant aerosol profile.

Technical comments

p.3, l.57, p.5, l.109, p.8, l.203, p.10, l.231, p.17, l.294: Please check a period.

---

## Author Comment (AC1) · 14 Jul 2020

**Response to Reviewers of:**
**"Using a coupled LES-aerosol radiation model to investigate urban haze: Sensitivity to aerosol loading and meteorological conditions" by Jessica Slater et al. 2020, submitted to ACP**

**General Response**

We thank the reviewers for their review and comments which will help to further improve the readability and understanding of this manuscript. Referee #1 states that including the aerosol radiation scheme in LES models improved understanding of aerosol-boundary layer interactions. The main concerns with this work focus on clarification of certain methods and choices made during the experimental setup including: aerosol size bins used in the model, how aerosol parameters were chosen for initialisation, the aerosol radiative properties included and why more periods weren't simulated. Referee #2 states that the use of the novel coupled LES-aerosol model provided more information on the sensitivity of PBL dynamics to aerosols. The referee suggests further discussion on the processes and interactions between aerosols, meteorology and PBL dynamics. They also suggest a more detailed explanation of the response of PBL dynamics to aerosol loading caused by meteorological conditions. Referee #3 finds the results presented in the manuscript to be interesting when compared to typical investigations using coarse-resolution models.

We have addressed the main problems addressed in the referee comments and specific responses are addressed below. Small technical problems including grammatical errors and mislabelling of figure captions pointed out by the referee have been corrected in the manuscript.

**Response to RC1**

*In experimental setup, please specify how the aerosol size distribution and volume fraction of chemical composition were determined. Were they adopted by in-situ measurements?*

The following line has been added to the manuscript: *"The values for aerosol size distribution data and composition fraction were taken from in situ measurements taken at IAP at 8am on 24th November 2016."*

*In addition, why not consider the impact of nitrate and ammonium on radiation transfer given that both of them contribute a large fraction of aerosol mass loading in Beijing.*

We thank the reviewer for this insightful comment and apologise for the mistake in the paper. Simulations presented in this paper included the impact of both nitrate and ammonium on radiative transfer, the line that claimed they weren't has been removed from the text and replaced with:
*"In all cases, BC can be considered to be the primary absorbing aerosol, with sulphate (SO4-), nitrate (NO3-) and ammonium (NH4+) strongly scattering and OC predominantly scattering with a small absorbing component."*

*Also, a description of aerosol size bins should be necessary. Like how many bins are separated and what are their ranges?*

A diagram of the size bins used in SALSA has now been added to the manuscript.

*The simulation was conducted for only three days. Why chose those days, were pollution or meteorology on 24-26 NOV. in 2016 very typical in Beijing during cold season?*

The aim of this work was 1) to examine the ability of UCLALES-SALSA to simulate the effect of aerosol-radiation interactions and their resultant feed back on meteorological conditions and 2) To assess the sensitivity of the aerosol-feedback loop to aerosol loading and initial meteorological conditions. This work took a haze episode in Beijing as an example to further understanding of these processes. In many ways the 24th-26th Nov 2016 can be considered typical of a haze episode in terms of the changing synoptic conditions which mark the haze episode beginning and end, and the stagnant meteorological conditions in the middle which are associated with rapidly increasing concentrations of aerosols, as they accumulate in a shallow boundary layer. However, the larger meteorological changes which can affect pollution episodes in Beijing, as well as the sources of pollution can vary during different episodes. These effects are not considered here and may have an impact on the magnitude of the effect as the meteorological conditions and aerosol composition loading will differ in different pollution episodes. However, this work just outlines the high sensitivity of the impact, examining specific pollution episodes in Beijing and how this effect differs is an idea for future work.

*Line 198: The constant concentration in the column may not be an appropriate representation of urban pollution since most emissions in the city located at surface (i.e., traffic). Especially the model is initialized at 8am in the morning when most pollutants are confined in a shallow near-surface layer.*

This is the case for the constant column concentration, however the aim of this paper is to show the sensitivities to such variations rather than fully simulate a case study of Beijng haze episodes (See above).

*Figure 6: Under the high aerosol loading situation, the entire layer of aerosol mass mixing ratio seems to decrease. Excluding the dry deposition why the column aerosol loading experienced a reduction?*

This is a very good point. When performing simulations with varied vertical aerosol profiles, there was a noticeable loss in mass of aerosols over time, despite dry deposition being switched off. The reason for the decrease in total mass mixing ratio over time is due to UCLALES-SALSA's use of the anelastic approximation. Anelastic models filter out acoustic waves for computational efficiency and are accurate in isentropic systems, which although useful, has limitations when applied to realistic atmospheric processes when the isentropic base state doesn't hold true.

The following explanation has now been added to the discussion:

*"It should be noted from the varied aerosol vertical profile simulations that total aerosol mass mixing ratio decreases by about 5 % over the course of the day. This is despite dry deposition not being included in these simulations. This is a result of UCLALES-SALSA using the Ogura-Philips anelastic approximation for filtering out acoustic waves. The approximation assumes that there are only small variations in pressure and density from static reference values over time. Throughout the day, surface fluxes increase air temperature, while subsidence of air at the model top decreases density (Ogura and Phillips, 1962; Pressel et al., 2015; Byun,1999). The limitations of the anelastic approximation mean that these changes do not fully feed back to change pressure, and fixed boundary conditions mean that volume remains constant. As the model holds to constant volume rather then constant mass, when SALSA aerosol mass tracers are pulled downward, the total air mass increases while the mass of aerosols remain the same, this causes the apparent decrease in aerosol mass mixing ratio (Figure 7). We consider this to be a limitation of using a meteorological model for air quality analysis, however as the relative reduction is the same for different meteorological conditions, comparisons can still be performed."*

**Response to RC2**

*2. Figure 2(b): Why did all cases underestimate the latent heat flux to a large extent?*

The latent heat flux calculation in the model is strongly affected by the water volume fraction at the surface. This value will change depending on conditions such as drainage and rainfall, but isn't a typically measured parameter in urban environments. The value used for water volume fraction was 0.3, sensitivity studies performed show that increasing this value, increased latent heat flux, however, this decreased sensible heat flux. As turbulence in Beijing is predominantly driven by sensible rather than latent heat flux, and the latent heat calculation in the model simulations are driven by a parameter which isn't well measured we consider this to be a limitation of our study.

*Why did you only simulate daytime but not the full day including night?*

Although stagnation overnight and a shallow boundary layer rapidly results in aerosol accumulation and can lead to further stagnation the following day. The idea behind the aerosol-PBL feedback is to show the interactions of aerosols with solar radiation which impact the pollution episode- the simulations do run into the night but overnight there isn't much effect. Due to the high computational cost of these simulations and the long night times in winter Beijing it wasn't feasible to run through the night

*Line 203: Why did you assume SO4 strongly scattering but ignore NO3?*

See above, NO3 and NH4 are included in the radiation scheme.

*4. Line 230: Although it is clear that aerosols have an effect on the decrease in the PBL height, the inclusion of high and low aerosol concentration has a different effect on the temperature in the lower and upper boundary layer as shown in Figure 4. For example, a low concentration of aerosol causes cooling in the upper layer, while the high concentration of aerosol causes warming there on 24/11; low concentration of aerosol causes slight cooling in the lower layer on 25/11, while cause slight warming on 26/11. I suggest adding some more detailed discussion instead of giving a rough conclusion as "inclusion of aerosols causes cooling in the lower boundary layer and warming in the upper layers".*

The following description has now been added to the discussion.

*"In all cases inclusion of aerosols causes cooling in the lower planetary boundary layer, and warming above it. This is due to the aerosols absorbing and scattering incoming SW radiation to reduce the amount of solar radiation reaching the surface. Where there are high concentrations of aerosols through the column, this severely reduces the amount of radiation reaching the surface and consequently causes cooling. Aerosols, specifically black carbon, in the upper layer of the boundary layer will absorb radiation, which causes warming. This effect is enhanced when aerosol concentrations are higher and this works to enhance temperature inversions and suppress PBL development."*

*Table 3: Could you add some discussion on the difference in the percentage of decrease in PBL height between high and low aerosol cases? The high concentration of aerosol causes a much larger decrease in PBL height on 26/11 than that on 24/11 and 25/11. This discussion would help to understand more about the effect of aerosol on PBL height and its influence factors. I saw you gave some explanation in Section 5.2, but I think those were still not enough. First, why high aerosols will cause >1 oC cooling on 26/11, while only 0.3°C on 24/11? Second, it looks like the high aerosols decrease the potential temperature ~1 oC in the lower layers on 25/11, but why PBL height was only decreased by 20%, which was much smaller than that on 26/11.*

This description has now been added to the manuscript.

*"For the case of 25/11, the PBL is already low due to synoptic conditions, and aerosols from the previous day causing strong temperature inversions in the morning. Therefore, even though the aerosols cause cooling in the PBL to the same amount on 26/11 and 25/11, a strong temperature inversion exists already on 25/11 and so the PBL development is low even without the inclusion of aerosols. "*

*Line 252: Varied aerosol vertical profiles minimize the increase in vertical velocity close to the model top, while they also increase the vertical velocity in the lower layers. This section (4.3) was talking about the effect of aerosol vertical profiles, but this paragraph only mentioned it with a half-sentence. The comparison in the vertical velocity is interesting and informative as shown in Figure 8. Please give more results about the effect of aerosol vertical profiles. Maybe separate the effects of high aerosol concentration and aerosol vertical profiles on vertical velocity into two sections.*

The following discussion has now been added to the manuscript

*"Reduced aerosol concentrations at high altitudes reduces vertical velocity at model top. This shows that aerosols at close to model top increase vertical velocity, through creating a turbulent layer. This is due to aerosol warming aloft close to model top causing stratification of the layer. Reduced aerosol concentrations in the entire column means that more solar radiation reaches the surface in the varied vertical aerosol profile case, increasing buoyant turbulence and vertical velocity in the PBL."*

*Line 273: Could you state more detail on how aerosol-radiation interactions of the previous night affect the meteorological condition on 25/11? Else, I suggest discussing more the sensitivity of boundary layer dynamics to meteorological conditions and its detail processes, instead of talking much about the variation of observed aerosols during these days.*

The following has now been added to section 5.1

"Aerosol-radiation interactions reduce the amount of solar radiation reaching the surface which causes cooling, simultaneously black carbon aerosols will absorb radiation at PBL top. Although absorption by black carbon (BC) occurs throughout the column, several studies have shown that due to the higher incidence of solar radiation and lower density of air, BC causes warming at PBL top to a greater extent than at the surface. Overall, this causes a temperature inversion during periods where pollution is high and causes a shallow PBL to form during the day. This leads to stagnant conditions and can affect the meteorology of the next day, particularly when aerosols are suppressed in a shallow PBL. However, frequently in Beijing wintertime, changes in pressure can cause warm polluted air to converge with cold clean air to create a layer of cold air under a layer of warm air. These conditions often pre-empt pollution episodes in Beijing and favour the accumulation of pollutants in a shallow boundary layer."

*Figure 7: Does the turquoise line mean the aerosol profile does not change (significantly) in the case with the constant aerosol profile? If yes, why not?*

In the case with the constant aerosol profile there is no change in the aerosol profile over time. As concentrations are the same through the column there is no difference in concentration and so mixing through diffusion vertically in the column does not occur. This has now been clarified in the figure legend.

*Line 317: This result was enough to prove the ability of your model to simulate the interaction between aerosol and meteorological conditions, but I would recommend adding more discussion on the detailed process of the interaction between aerosol and meteorological condition within the PBL using your model.*

We are not quite sure necessarily what description the reviewer is referring to here, we think that this is just a further description of how the aerosols perturb PBL meteorology. The following has been added as a further description in section 5.2.

" In these simulations, the aerosols interact with radiation to cause heating and cooling in different layers which perturbs the temperature profile of the PBL and decreases the sensible heat flux term. The aerosols also take up water to a small extent which decreases latent heat. These effects lead to decreased turbulence in the PBL, when aerosol concentrations are high."

*Line 125: what are the refractive indexes you used in the model? Please add a table or reference*

The refractive indices are based on Bond and Bergstrom (2006) a table including refractive indices used has been added to the manuscript for information.

*Figure 2: The potential temperatures at 8 pm are shown on the left, but you did not simulate anything at 8 pm. As you stated, the model ends at 6 pm each day. Why did*

*you show the temperature at 8 pm here?*

Simulations were performed from 8am until 10pm.

*Figure 5: Why did not use a unit of mixing ratio in Figure 5 like Figure 6&7? Making them the same will be better to understand.*

Although the authors recognise that having the same units on all figures may make for a better comparison, the mixing ratio which would be presented in figure 5 is the same as for the constant profile in figure 6. As this paper is associated with Beijing air quality where aerosol mass concentrations are in µg/m³ and this has been discussed in the text, figure 5 provides an indication of how mass mixing ratios are associated with mass concentrations which has implications for health.

All below comments have been either amended or clarified in the manuscript

*Line 23: It will be better to add some references for these observation results.*
*Line 32: Change distribution to diffusion?* Changed to vertical mixing
*Figure 5: The color in the legend is wrong.* This was an issue in the final subplot, which has now been changed.

*12. Figure 6: where are dashed red lines? Could you make them more obvious? Or give some explanation in the legend.* This has now been explained in the figure caption.
*14. Figure 7: where is the blue solid line?* This has now been explained in the figure caption.
*15. Figure 8: I suggest adding a short subtitle (like date & case) for each subplot to make readers easy to understand.*
*19. Line 315: What do a) and b) stand for here? I didn't find a subplot in Figure 7* This has now been better clarified in the text.

**Response to RC3**

*p.4, l.83: Please define cloud and fog in the model. How did you consider the interaction with aerosols?*

Although cloud/fog activation and interactions with aerosols are available in the model, they are not considered at any point during these simulations. Primarily this is because humidity is too low, that even at high aerosol concentrations, cloud and fog do not form.

*p.5, l.127: How did you consider the lens effect?*
The lens effect was not considered in this work. Rather, although all aerosols were internally mixed, the refractive indices were calculated based on the volume-weighted average of all chemical components in each size bin.

*p.8, l.196: How did you set up the cloudless with varying aerosol concentrations?*
The humidity was not high enough for clouds to form in any of the simulations.

*p.8, l.201: Please specify the small absorbing component of OC*
A table with refractive indices for SW radiation has now been included in the text.

*p.8, l.203: How did you consider the gravitational setting of aerosols? Please explain the boundary condition for aerosols.*

In this work there was no net flux of aerosols at the boundaries, i.e although there is capability in UCLALES-SALSA, neither emission and deposition of aerosols were considered in these simulations.

**References**

Byun, D. W.: Dynamically consistent formulations in meteorological and air quality models for multiscale atmospheric stud-ies. Part II: Mass conservation issues, Journal of the Atmospheric Sciences, 56, 3808–3820, https://doi.org/10.1175/1520-0469(1999)056<3808:DCFIMA>2.0.CO;2, 1999

Ogura, Y. and Phillips, N. A.: Scale Analysis of Deep and Shallow Convection in the Atmosphere, https://doi.org/10.1175/1520-0469(1962)019<0173:saodas>2.0.co;2, 1962.

Pressel, K., Kaul, C., Schneider, T., Tan, Z., and Mishra, S.: Large eddy simulation in an anelastic framework with closed water and entropy balances, Journal of Advances in Modeling Earth Systems, 7, 1425–1456, https://doi.org/10.1002/2017MS001065, 2015